# Evaluation of the PLA-nZH-Cu Nanocomposite Film on the Micro-Biological, Organoleptic and Physicochemical Qualities of Packed Chicken Meat

**DOI:** 10.3390/foods11040546

**Published:** 2022-02-14

**Authors:** Judith Vergara-Figueroa, Fabiola Cerda-Leal, Serguei Alejandro-Martín, William Gacitúa

**Affiliations:** 1Center for Biomaterials and Nanotechnology, Universidad del Bío-Bío, Concepción 4030000, Chile; wgacitua@ubiobio.cl; 2Wood Engineering Department, Faculty of Engineering, Universidad del Bío-Bío, Concepción 4030000, Chile; 3Nanomaterials and Catalysts for Sustainable Processes Group (NanoCatpPS), Universidad del Bío-Bío, Concepción 4030000, Chile; 4Food Engineering Department, Faculty of Health and Food Sciences, Universidad del Bío-Bío, Chillan 3780000, Chile; fcerda@ubiobio.cl

**Keywords:** nanocomposites, copper ions, chicken breast meat, food safety, foodborne bacteria, physicochemical quality

## Abstract

This research evaluated the contribution of nanocomposite films based on different concentrations of nZH-Cu (1%, 2%, and 3%) to the microbiological, organoleptic, and physicochemical characteristics of packed chicken breast meat. Analysis of some meat quality traits, such as microbiological, chemical, and physical, were conducted on a laboratory scale. For this, small squares of chicken breast meat, weighing approximately 10 g, were aseptically wrapped with rectangular pieces of 5 × 10 cm PLA-nZH-Cu nanocomposite films, which were stored at 4 °C for 20 days. The microbiological results indicated efficient antibacterial activity (at any nZH-Cu concentration in the nanocomposite films) on the total viable count of groups of psychrophiles, aerobic mesophiles, Enterobacteriaceae, and *Salmonella* spp. until day 10 of storage (*p* < 0.05). No significant changes were observed in the organoleptic (color) and physicochemical qualities (texture, weight, pH, and acidity) until day 10 of storage at 4 °C (*p* < 0.05). The analysis of the experimental tests carried out determined that the PLA-nZH-Cu nanocomposite films played an effective role in the bacterial safety of the packaged chicken. It was concluded that the nZH-Cu nanocomposite films, at all concentrations tested, extended the shelf life of the chicken breast meat for up to 10 days in a refrigerator at 4 °C.

## 1. Introduction

One of the main problems that food science and technology must address daily is microbial pathogens in food. These pathogens pose a threat to health, at least for some people. Many diseases are self-limited diarrheal syndromes in healthy humans since the pathogen is acquired through the digestive tract. However, some can cause life-threatening diseases that require rapid and specific antimicrobial therapy. For this reason, the food industry is constantly developing and implementing new procedures to minimize the possibility that even a single pathogenic microbial cell will survive in food. The problem worsens because these pathogens can survive the chemical or physicochemical procedures currently applied in the food industry to extend the shelf life of food [1].

According to White et al., (2002), in the United States, some microbial pathogens are responsible for approximately 95% of food-related deaths, where 34% of this figure corresponds to the Gram (−) anaerobic facultative fermentative *Salmonella* strains [2]. There have been countless outbreaks of *Salmonella* spp. throughout the Chilean territory, mainly in schools and public places, either due to poor food handling [3] or because the meat consumed was contaminated [4]. Outbreaks of *Salmonella* spp. have also been recorded due to eating raw eggs [5].

Chicken meat is among the most consumed food of animal origin globally in diverse cultures, traditions, and religions [6]. At the same time, it is one of the most perishable foods present in the food industry or commerce. Meat represents an ideal medium for the growth of pathogenic or spoilage microorganisms. Most healthy animal tissues are known to be germ-free [7]; however, during meat handling and processing, microorganisms can gain access to the meat from its the surfaces. In addition to the development of pathogens, microbial growth commonly induces undesirable organoleptic changes during meat storage. Food spoilage results from microbial activity in the food matrix that causes the breakdown of carbohydrates and proteins [8,9,10]. Meat and meat products are commonly contaminated with *Listeria monocytogenes*, *Salmonella typhimurium*, *Salmonella enteritidis*, and *Yersinia enterocolitica*. *Salmonella* spp. is one of the leading causes of bacterial foodborne diseases in developed and developing countries [9]. According to the United States Government, Department of Agriculture (2013), classic packaging protects chicken meat for 1 to 2 days under refrigerated conditions [11] and 3 to 5 days, according to an article reported elsewhere [12].

The main groups of bacteria used as indicator microorganisms for meat quality are the psychrophilic, aerobic mesophilic, Enterobacteriaceae, and *Salmonella* spp. [13]. Moreover, these microorganisms can be used to estimate the shelf life of food. Thus, counts between 7–8 CFU/mL Log_10_ indicate product alterations that make the product unsuitable for human consumption [14,15,16,17]. Table 1 shows the characteristics of the meat quality indicator bacteria.

The bacteria mentioned above have defined characteristics, which make them sensitive to the action of different types of bactericides, including copper ions (Cu^2+^). Furthermore, the Gram (+) bacteria have a thick peptidoglycan layer and the Gram (−) have a lower ratio peptidoglycan layer and an outer lipopolysaccharides membrane. A lower proportion of peptidoglycan and the negative charge associated with lipopolysaccharides could have a higher affinity for positively charged ions. This interaction would allow a more significant association and incorporation of ions through the outer membrane of Gram (−) bacteria, with the consequent cellular damage described in the literature (Figure 1). However, these effects depend on the type of microorganism, type of compound, physical or chemical factors, concentrations, and physiological state of the bacteria [18,19].

Considering the preceding information, it is necessary to guarantee the safety of food products. There is currently an emphasis on food safety caused by cross contamination, which causes spoilage by microorganisms found on the surface of other foods [20,21,22,23]. A potential alternative to preventing the development of pathogenic microorganisms and the deterioration of meat food products is antimicrobial packaging that is mainly based on biodegradable polymers and natural components [20,24,25,26,27,28].

The use of nanotechnology in food packaging is an emerging area in which the new packaging materials can be manipulated to improve their new capabilities. Moreover, the increase in barrier quality, mechanical properties, heat resistance, and biodegradability means that the packaging containers are endowed with antimicrobial qualities. Incorporating antimicrobial compounds in the packaging matrix could support a functional effect on the food surface [20,29,30,31].

In this context, polylactic acid (PLA) is a natural, biodegradable, biocompatible, and non-toxic polymer [32,33,34]. PLA is an electrospinning process that can produce fibers. These fibers can form films for applications such as food wrappers, which have attributes according to the required needs. Several investigations have highlighted its versatility and its ability to form hydrophobic biocomposites, which increase the mechanical strength of the film and its ability to encapsulate and control the release of nanocomposites [25,26,33,34,35,36,37,38,39,40]. Electrospinning of fibers loaded with zeolite enables catalytic, antibacterial, filtration, and medical-use membrane properties [41,42,43,44,45,46]. Natural nanozeolite particles exchanged with copper ions (Cu^2+^) can be incorporated directly into a food contact film, allowing ion retention at the zeolite framework. The ions must be kept within the zeolite structure in order to perform their antimicrobial activity. Natural zeolite is not a practical antibacterial material but it can function as a carrier for ions that can be used as an antibacterial agent. Ion exchange is carried out by replacing the sodium (compensating cation) in the zeolite structure with other ions, such as copper [47].

Nanoparticles made of copper oxides are non-expensive, non-toxic to humans, abundant, and have stable physical and chemical properties. These properties give them advantages over other metal oxides and make them usable worldwide. Furthermore, they are easy to mix with polymers and other matrices and can be prepared with various surface morphologies [48]. Therefore, the antimicrobial activity of metal oxide nanostructures could apply in the food industry. In addition, the nanometric dimensions of these particle and their large specific surface allow them to interact with bacteria. The main theories on the bactericidal effects of nanostructured metal oxides and ions (Figure 1) are oxidative stress and the formation of reactive oxygen species (ROS); cell wall-membrane damage due to electrostatic interaction and accumulation; loss of homeostasis due to metal ions; protein, and enzyme dysfunction; genotoxicity, and inhibition of signal transduction [19].

The cations present on the surface of the nanoparticles formed by metal oxides are Lewis acids with unsaturated valences that can form dative covalent donor bonds, with Lewis bases (nitrogenous and oxygenated groups) present in the amino acids and polysaccharides in the cell walls [48]. This interaction can also generate lysis of the cell walls [19].

Although active packaging provides meat protection against pathogenic microorganisms, it is essential to evaluate its influence on the quality of the contained meat [16,49]. In this context, this research aimed to determine the microbiological, organoleptic, and physicochemical qualities of chicken breast meat wrapped with rectangular pieces of PLA-nZH-Cu nanocomposite films. The tests were carried out at 4 °C and stored for 20 days. The films were manufactured using the dual configuration electrospinning technique described in a previous study [24].

## 2. Materials and Methods

### 2.1. Materials

#### 2.1.1. nZH-Cu Nanocomposite Films

Nanocomposite films with different concentrations of nZH-Cu (1%, 2%, and 3%) were obtained following a procedure reported in previous studies written by the same authors [24,50]. The abbrieviation nZH-Cu (1%, 2%, and 3%) stands for nanoparticles of Chilean natural zeolite with ion exchange and copper salt (Cu^2+^) in different concentrations [50].

Previously, nanocomposite films were subjected to morphological analysis, evaluation of their physical–mechanical properties, thermal degradation profile, and permeability to water vapor [24]. The nanocomposite film that presented the best mechanical, physical, and barrier properties was selected to be applied in the current investigation [24]. From this, the influence of the different concentrations of nZH-Cu in the film (1%, 2%, and 3%) on the microbiological, organoleptic, and physicochemical qualities of the packaged chicken meat was evaluated. As a control, films of pure PLA and PLA-nZH (3%) were used. nZH corresponds to zeolite nanoparticles with increased surface areas due to the cleaning of their pores and channels [50]. Figure 2 shows a prototype of the PLA-nZH-Cu nanocomposite films.

#### 2.1.2. Chicken Breast Meat

Unmarinated commercial chicken breast meat samples were used in the tests to not interfere with the microbial load of the meat (these were branded breast fillets that were individually frozen).

#### 2.1.3. Reagents and Equipment

Phosphate buffer sterile solution was used to wash the chicken breast meat and dilute the samples. Sodium hydroxide at 0.1 N was used to determine the total titratable acidity. The following types of agar were used: plate count, EMB- LEVINE, Hektoen, agar–agar, and TSI. All reagents were purchased from Merck (Darmstadt, Germany).

A pH meter (Hanna, Instruments, Woonsocket, RI, USA) was used in the pH measurement. A texturometer (TA-XT Plus Stable Micro Systems, Godalming, UK) evaluated the texture with a Warner-Bratzler (Godalming, UK) blade accessory. The instrumental color evaluation was conducted using a colorimeter (Konica Minolta, Valencia, España, model CR 400) with an illuminant D65 10° observer. The sample was crushed with an Ultraturrax homogenizer (Heidolph model T25 Basic, Wilmington, NC, USA) before acidity determination.

### 2.2. Methods

The trial methodology was adapted from the one described by Castañeda et al., (2013) and Machado de Melo et al., (2012) [16,51]. The main objective was to evaluate the influence exerted by nanocomposite films with different concentrations of nZH-Cu (1%, 2%, and 3%) [24,50] on the microbiological, organoleptic, and physicochemical characteristics of chicken breast meat.

Initially, the chicken breast meat was cut into small squares of approximately 10 g and finally wrapped aseptically with rectangular pieces of 5 × 10 cm PLA-nZH-Cu nanocomposite films. The wrapped chicken pieces were then placed inside sterile bags and stored at 4 °C. Finally, the test was conducted at 4 °C as reported in “Food Handling Recommendations” [52]. 

For all evaluations, samples were analyzed in triplicate after 0, 10, and 20 days of storage. The description of the steps carried out during days of storage can be seen in Figure 3.

#### 2.2.1. Evaluation of the PLA-nZH-Cu Nanocomposite Film’s Role in the Microbiological Quality of Chicken Breast: Total Viable Count of Psychrophiles, Mesophilic Aerobes, Enterobacteriaceae, and *Salmonella* spp.

For the microbiological analysis, viable counts of psychrophilic, aerobic mesophilic, enterobacteria, and *Salmonella* spp. bacteria were considered [13]. First, approximately 10 g of the chicken samples were taken and homogenized for 2 min in a sterile bag with 90 mL of phosphate buffer; a Stomacher 400 Circulator (Seward Laboratory, London, UK), was used for 2 min to perform this. Next, decimal serial dilutions were aseptically made with the same buffer solution from which samples were taken to be incubated on different types of agar.

For the count of psychrophilic and mesophilic bacteria, 1 mL of each dilution was taken and poured into a sterile Petri dish with 20 mL of plate count agar. Plates were incubated at 10 °C for ten days for psychrophilic counts and at 30 °C for three days for mesophilic counts.

For the enterobacteria count, 1 mL of each dilution was taken and poured into Petri dishes, and counts were determined using EMB Levine agar and Hektoen agar. The plates were then incubated at 30 °C for 48 h.

The description of the steps carried out during the test can be seen in Figure 4.

#### 2.2.2. Evaluation of the PLA-nZH-Cu Nanocomposite Film’s Role in the Organoleptic Quality of Chicken Breast: Instrumental Color (Chroma, Hue, and E)

With the help of the Konica Minolta equipment, the color changes in the chicken samples were evaluated during the 20 days of storage. Hue, Chroma, and E parameters were determined according to Mathias Rettig and Ah-Hen, using the “Visual Method, Colorimeter” [53,54,55]. The description of the steps carried out during the test can be seen in Figure 5.

#### 2.2.3. Evaluation of the PLA-nZH-Cu Nanocomposites Film’s Role in the Physicochemical Quality of Chicken Breast: Texture, Weight, pH, and Acidity

Physicochemical qualities, firmness, weight, pH, and acidity during the 20 days of storage were considered. In addition, the texture of the chicken breast meat was evaluated using Stable Micro Systems equipment (TA-XT Plus Stable Micro Systems, Godalming, UK). The firmness (N) parameter was obtained according to Braña et al., 2011 using the “Laminar Cut Method” [56]. The description of the steps carried out during the test can be seen in Figure 5.

### 2.3. Statistical Analysis (ANOVA)

The trials were performed in triplicate with a 95% confidence level and randomized experimental design. ANOVA and the multiple comparison test were performed in the Statgraphics Centurion XVI program (16.1.03, Statgraphics Technologies, Inc., The Plains, VA, USA).

## 3. Results and Discussion

### 3.1. Evaluation of the PLA-nZH-Cu Nanocomposite Film’s Role in the Microbiological Quality of Chicken Breast: Total Viable Count of Psychrophilic, Mesophilic Aerobes, Enterobacteriaceae, and Salmonella spp.

The total viable counts of the psychrophiles, mesophilic aerobes, Enterobacteriaceae, and *Salmonella* spp. are primary indicators for the microbiological quality of meat and can be used to estimate the shelf life of food [13]. Therefore, the food industry considers that the microbiological load of meat is an important parameter to examine before its commercialization [15,57,58]. Previous studies suggested that food with a total bacterial load between 7–8 CFU/mL Log_10_ was unsuitable for human consumption [14,15,16].

The psychrophilic bacteria group (Table 2) increased from 2.36 to 5.80 CFU/mL Log_10_ in the control after ten days of storage. There was less growth in all treatments for chicken meat samples wrapped with PLA-nZH-Cu nanocomposite films (*p* < 0.05). The samples of chicken breast meat that showed a lower growth of psychrophilic bacteria were wrapped with the 1% and 3% PLA-nZH-Cu nanocomposite films. The same tendency for this bacterial group was observed after 20 days of storage, with a more significant increase in psychrophilic bacteria for the control (8.47 CFU/mL Log_10_) and less for the samples wrapped in PLA-nZH-Cu nanocomposite films (Table 2). The samples that showed a lower psychrophilic bacteria growth were those wrapped with the 1% and 3% PLA-nZH-Cu nanocomposite films. A higher total viable count was observed for the group of psychrophilic bacteria compared to the other bacterial groups evaluated. This increase may have been related to the fact that psychrophilic bacteria have a more remarkable adaptation to low temperatures, such as refrigeration temperatures (4 °C). For this reason, their count could have been higher than the other bacterial groups.

For the group of mesophilic aerobic bacteria, a higher increase was observed for the control sample after 10 days of storage (3.10 to 4.54 CFU/mL Log_10_). The wrapped samples showed lower bacterial growth than the control samples. The treatment that showed the lowest bacterial growth was the one wrapped with the 3% nanocomposite films of PLA-nZH-Cu. However, after 20 days of storage, there was a high growth of this group of microorganisms (Table 2). The samples in contact with the films indicated that growth was slightly lower than the control. After 20 days of storage, the sample that showed the lowest bacterial growth was wrapped with the 2% nanocomposite films PLA-nZH-Cu.

For the enterobacterial group, moderate growth was observed after 10 days of storage. There was a significant decrease in the growth of this group of bacteria when using the 3% nanocomposite films PLA-nZH-Cu. After 20 days of storage, a considerable increase in this bacterial count was observed. The samples with the 3% nanocomposite films PLA-nZH-Cu had less growth than the control samples (Table 2). Within the group of Enterobacteriaceae, there are several bacteria that are psychrophilic. The importance of detecting and quantifying them lies in the fact that their presence has been described in refrigerated meats, which could cause the deterioration of these foods [13,14,16].

Regarding the growth of *Salmonella* spp., it was indicated that there was no growth during the test days for all chicken breast samples.

In comparison with other studies, it was indicated that the results reported here were superior to those reported by [16] Mechado de Melo et al., (2012). However, the results of this study were lower than those reported by [49] Ahmed et al., (2018). The results of this research indicated that PLA films with inserted nanometals (Ag-Cu) and essential oils showed sustained control of *Campylobacter jejuni*, enteric *Salmonella*, *Typhimurium*, and *Listeria monocytogenes*. In this research they used antibacterial agents in the film that was in contact with the chicken, which, after six days of treatment, had a growth of psychrophile bacteria of 7.71 CFU/mL Log_10_ [16]. However, after 150 days of the test, there was a growth of 5 CFU/mL Log_10_ of this bacteria [49]. The bimetallic action and the action of the active principles of the essential oils possibly exerted a synergistic effect in order to eliminate the bacteria.

There are several mechanisms and theories on the bactericidal effect of supported metal ions in nanostructures [24,29,30,32]. Antimicrobial activity can be associated with nanometric dimensions and a large specific surface, which allows it to interact with bacteria [19]. It has been suggested that the formation of reactive oxygen species can produce harmful effects in cells such as damage to its DNA and damage due to the oxidation of polyunsaturated fatty acids and amino acids, which causes oxidative stress in the cells. On the other hand, polysaccharides in bacterial membranes have electronegative groups at the sites of attraction in metal cations. The difference in charge between them causes electrostatic attraction; they accumulate on the surface of the bacteria and alter the structure and permeability of the cell membranes. Gram (−) bacteria have a higher negative charge than Gram (+) bacteria. Therefore, the electrostatic interaction is stronger in Gram (−) strains [24,50]. Bacterial membranes have pores of nanometric sizes, so the smaller the bactericidal particle, the larger its surface area will be, which leads to greater bactericidal efficacy. Another theory indicates that protein and enzyme dysfunction can occur, which is caused by protein carbonylation; this leads to the loss of catalytic activity in the case of enzymes, which ultimately triggers the breakdown of proteins and can cause cell destruction. A different explanation is that due to their electrical properties, metal oxides and ions interact with nucleic acids, in particular genomic and plasmid DNA. They suppress the cell division in microbes by altering the replication processes of chromosomal DNA and plasmids, so signal transduction in the bacteria is affected [48].

When comparing the PLA and PLA-nZH films with the PLA-nZH-Cu nanocomposite films, it was observed that in some cases the PLA and PLA-nZH films presented bacterial count values similar to the films with nZH-Cu. It should be noted that PLA films by themselves do not have antibacterial characteristics [1,34]. Possibly, and unintentionally, these films could have been in contact with the nanoparticles, which probably could have caused the films to be loaded with nZH-Cu particles. This may be the explanation that, in some results, the film’s control presented bacterial count values similar to the PLA films with nZH-Cu. Furthermore, and according to the results, it could be assumed that there was an adaptation period for a different group of bacteria during storage. Subsequently, an increase in bacterial growth rate was observed, mainly for psychrophilic bacteria [16,49]. The increase observed in the present investigation was probably due to the growth of the superficial microbiota of the chicken pieces that were not in contact with the films [18,19].

Therefore, one of the factors that influenced the bactericidal action of the films was the food matrix. The interaction of the different components, both nutritious and non-nutritive, gave the meat specific physicochemical characteristics. These characteristics could create a suitable or hostile environment for the meat against the action of antibacterial agents [15]. According to the present investigation, the films of PLA-nZH-Cu nanocomposites reduce the growth of the bacterial load in the chicken pieces compared to the control samples. Furthermore, it was evident that this behavior was dependent on the concentration of nZH-Cu in the films (Table 2). Therefore, it could be established that nanocomposite films effectively provided adequate bacterial conditions for up to 10 days in refrigeration conditions at 4 °C. However, after 20 days, under the same conditions, the chicken breast meat would not be safe for human consumption [14,15,16,17].

An explanation for these results can be found in the fact that the films proposed in this research (Figure 6) showed a rapid migration of their antibacterial action. The antibacterial effect was not maintained throughout the 20 days of testing in the refrigerated state. If an antimicrobial could be released from the package during an extended period, the activity could also be extended into the transport and storage phases of food distribution [10].

### 3.2. Evaluation of the PLA-nZH-Cu Nanocomposite Film’s Role in the Organoleptic Quality of Chicken Breast: Instrumental Color (Chroma, Hue, and E)

The chroma, hue, and E parameters were not strongly affected during the 20 days of storage. They remained without significant changes until day 10 of storage (Table 3). This meant that the chromaticity (saturation, intensity, and purity) of the samples wrapped with PLA-nZH-Cu nanocomposite films did not change in color grade. Furthermore, the samples’ hues (tint and color) did not show any wavelength changes due to radiation. Regarding E (quantification of color change) it was indicated that, when comparing one sample with another, a slight increase in the color change could be observed after 10 days of storage. These results indicated that it was likely that neither lipid oxidation nor metmyoglobin formation were evident during this period [16]. In addition, as shown in Table 4, the results of the physicochemical parameters also did not see their values modified until 10 days of storage.

It can be stated that PLA-nZH-Cu nanocomposite films did not negatively affect the color parameters (chroma, hue, and E) in the chicken samples. These results may indicate that PLA-nZH-Cu nanocomposite films could cause light scattering due to the incorporation of nanoparticles in the polymeric matrix. This quality could be advantageous when considering them as packaging films for food products sensitive to light and UV rays [16,49].

### 3.3. Evaluation of the PLA-nZH-Cu Nanocomposite Film’s Role in the Physicochemical Quality of Chicken Breast: Texture, Weight, pH, and Acidity

The physicochemical parameters of firmness, weight, acidity, and pH were slightly affected during the 20 days of storage (Table 4).

The firmness parameter showed an increase from 10 days of storage. In general, this increase was due to the hydrolysis of the proteins and degradative processes of the meat. It could also be related to dehydration of the sample caused by the passage of days of storage [16]. As the days of storage elapsed, there was an increase in the loss of weight in the sample. This result was more evident in the samples that were in contact with the pure PLA film, which was possibly caused by water displacement induced by the hydrophobicity of the PLA in the sample. High standard deviation values were observed for firmness (N), which may have been related to the characteristics of the sample. The state and interaction of the different muscle structures and their components (myofibrils, connective tissue, and water) could cause different behaviors in the sample during the treatments. The causes that gave rise to the variation in the firmness of the meat could be species, breed, production system, refrigeration and freezing system, meat maturation, shortening of sarcomeres (state of muscle contraction), quantity, and characteristics of the connective tissue [56,58,59].

As the days of storage elapsed, a slight decrease in lactic acid was observed in the sample (Table 4). At the same time, this caused a slight increase in the pH of the sample. Both differences were not significant. The decrease in lactic acid in the chicken breast accounted for the poor quality of the meat [16]. Lactic acid in the muscle slows the growth of bacteria that can contaminate the chicken breast meat. This decrease was caused by biochemical processes that occur in animal meat after the animal dies, which increases the pH. A pH of approximately 6.2–6.4 is considered acceptable in chicken meat [14]. Chicken meat stored under oxygenated conditions, which is rich in proteins and free amino acids, commonly presents an increase in pH as the number of microorganisms increases, which is responsible for the deterioration of the meat. In addition, its proteolytic activity produces essential compounds and causes a slight increase in pH [16]. Therefore, it was established that PLA-nZH-Cu nanocomposite films did not significantly affect the acidity and pH of the chicken samples.

The organoleptic and physicochemical characteristics of a food are used mainly as quality control and to distinguish its freshness [56,58,59]. For example, according to the USDA, chicken meat can last a maximum of 2 days in a refrigerated state [11]. Therefore, its consumption after that time is not recommended.

Thus, this study showed that the nanocomposite films investigated here were efficient in controlling the bacterial microbiota. Furthermore, they did not modify the organoleptic and physicochemical characteristics of the samples up to 10 days of storage in refrigerated conditions at 4 °C. These results are significant in fresh meats since their shelf life and safety in refrigerated conditions are diminished by the intrinsic growth of the main groups of bacteria, which indicate the quality of the meat (psychrophiles, mesophile aerobes, Enterobacteriaceae, and *Salmonella* spp.) [13].

Following the “Norma General Técnica Sobre Inspección médico Veterinaria de Aves de corral y de sus Carnes” 2010, Chile, the nanocomposite films evaluated in this research met the requirements requested by the Ministry of Health of Chile [59]. Moreover, under this standard, the PLA-nZH-Cu nanocomposite films evaluated here could be considered as clean biomaterials, which are free of polluting substances and odors that may cause alterations in chicken meat [59].

## 4. Conclusions

The role of PLA-nZH-Cu nanocomposite films on the microbiological, organoleptic, and physicochemical qualities of packaged chicken was evaluated here. The PLA-nZH-Cu nanocomposite films reduced the growth of the bacterial load in the chicken samples compared that of the control samples. The best effects were achieved using the film containing 3% nZH-Cu. Furthermore, after ten days, wrapped chicken meat samples (stored at 4 °C) did not significantly vary with respect to their organoleptic and physicochemical qualities; thus, the nanocomposite films contributed to the control of microorganisms without modifying the organoleptic and physicochemical characteristics of the meat. Hence, the shelf life of packaged chicken would be extended up to 10 days at 4 °C using the above-mentioned films.

## Figures and Tables

**Figure 1 foods-11-00546-f001:**
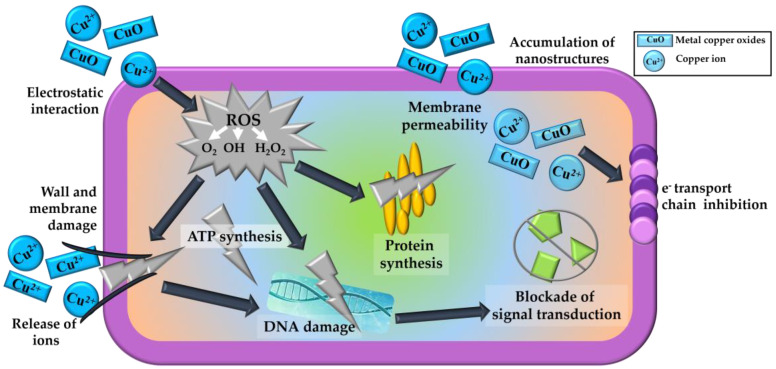
Scheme of theories of the bactericidal effects of nanostructured metal oxides and ions. Adapted from Vázquez-Olmos et al., 2018 [19].

**Figure 2 foods-11-00546-f002:**
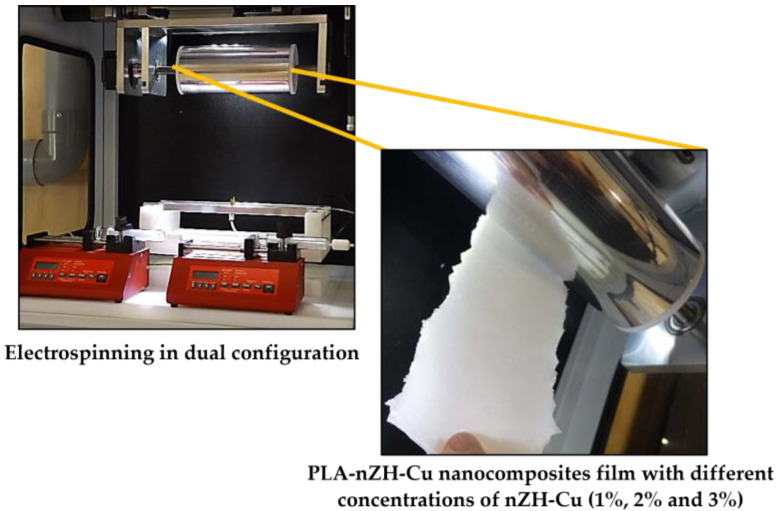
Nanocomposite films developed by the electrospinning technique in dual configuration with different concentrations of nZH-Cu (1%, 2%, and 3%). These nanocomposite films were used to wrap chicken breast meat.

**Figure 3 foods-11-00546-f003:**
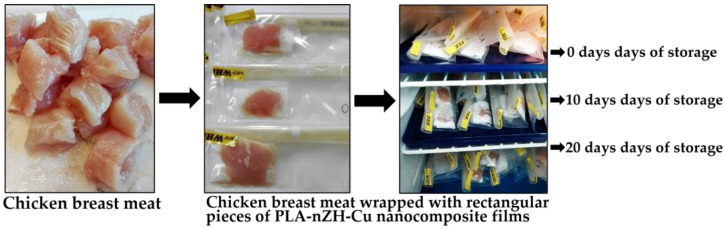
Chicken meat pieces wrapped in PLA-nZH-Cu nanocomposite films placed inside sterile bags and stored at 4 °C. The samples were analyzed at 0, 10, and 20 days of storage.

**Figure 4 foods-11-00546-f004:**
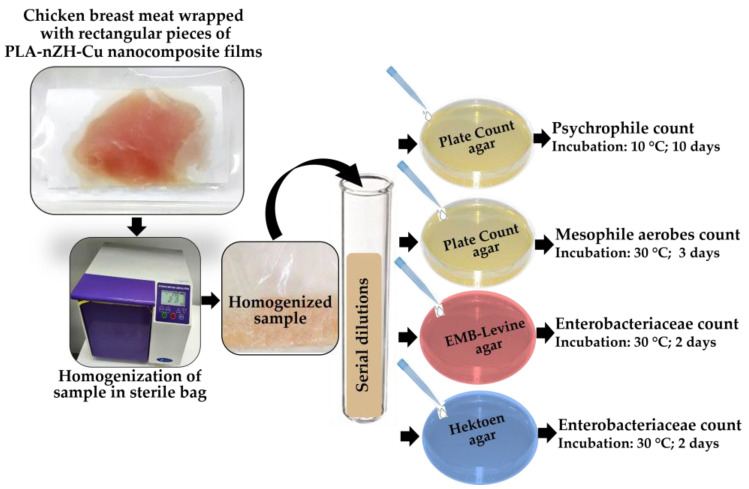
Evaluation of the PLA-nZH-Cu nanocomposite film’s role in the microbiological quality of chicken breast. Each test was carried out consecutively for 0, 10, and 20 days of storage.

**Figure 5 foods-11-00546-f005:**
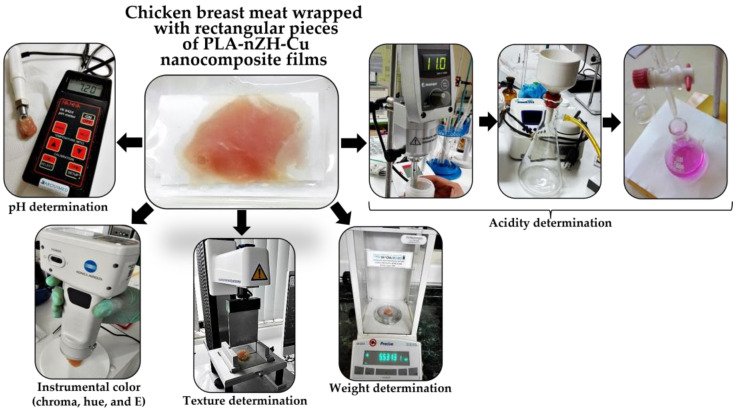
Evaluation of the PLA-nZH-Cu nanocomposite film’s role in the organoleptic and physicochemical qualities of the chicken breast. Each test was carried out consecutively for 0, 10, and 20 days of storage.

**Figure 6 foods-11-00546-f006:**
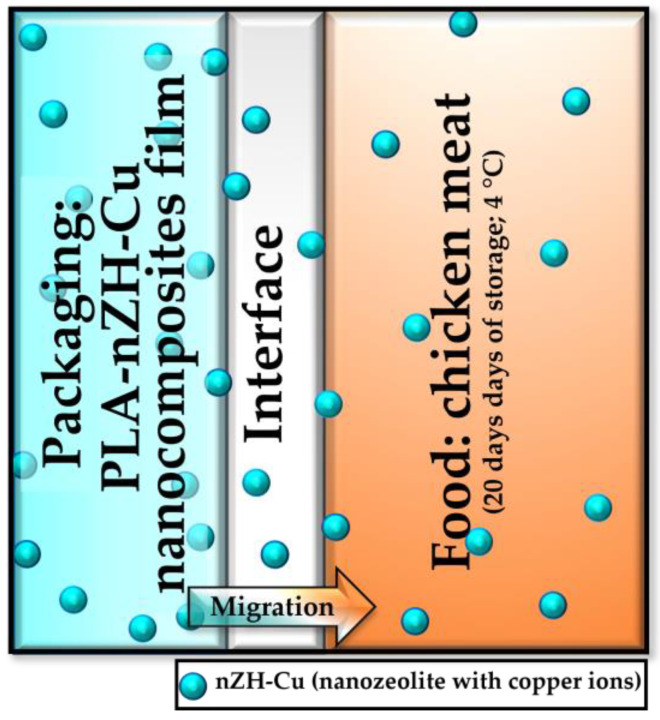
Scheme of the migration process, when the PLA-nZH-Cu nanocomposite film was in contact with the chicken meat. Adapted from Barros-Velázquez (2016) [1].

**Table 1 foods-11-00546-t001:** Meat quality indicator bacteria.

Bacteria Group	Characteristics
Psychrophile	The optimal growth temperature of psychrophiles is low, and they can reproduce in refrigeration conditions (4 °C). They are the main bacteria responsible for the deterioration of refrigerated foods of animal origin.
Mesophile aerobes	Mesophile aerobes have intermediate thermal characteristics. Their presence shows the sanitary and hygienic quality in the elaboration of food.
Enterobacteriaceae	Such microorganisms are found in the intestine and feces of animals, birds, and man. As they do not resist high temperatures they survive under refrigeration and drying conditions. Their presence in food accounts for the lack of hygiene and rigorous handling of food due to cross contamination.
*Salmonella* spp.	Enterobacteriaceae usually contaminate raw meat, poultry, eggs, and dairy products. *Salmonella* spp. can sometimes cause serious infections in young children, the elderly, and people with weakened immune systems.

Source: [13,14,16].

**Table 2 foods-11-00546-t002:** Microbial count (CFU/mL Log10) for chicken breast meat in contact with PLA-nZH-Cu nanocomposite films during storage at 4 °C for 20 days.

Microorganism	Treatment	Number of Days of Storage		
		0	10	20
Psychrophiles	Control	2.36 ± 0.13	5.80 ± 0.04 ^a^	8.47 ± 0.10 ^a^
	PLA	-	4.87 ± 0.02 ^b^	8.35 ± 0.13 ^b^
	PLA-nZH (3%)	-	4.82 ± 0.02 ^c^	7.90 ± 0.09 ^d^
	PLA-nZH-Cu (1%)	-	4.74 ± 0.08 ^d^	7.75 ± 0.21 ^e^
	PLA-nZH-Cu (2%)	-	4.91 ± 0.54 ^b^	8.27 ± 0.13 ^c^
	PLA-nZH-Cu (3%)	-	4.79 ± 0.01 ^c^	7.70 ± 0.02 ^d^
Mesophile aerobes	Control	3.10 ± 0.13	4.54 ± 0.06 ^a^	9.05 ± 0.04 ^a^
	PLA	-	4.34 ± 0.30 ^c^	8.70 ± 0.01 ^c^
	PLA-nZH (3%)	-	4.12 ± 0.06 ^d^	8.92 ± 0.12 ^b^
	PLA-nZH-Cu (1%)	-	4.27 ± 0.11 ^c^	8.87 ± 0.04 ^b^
	PLA-nZH-Cu (2%)	-	4.48 ± 0.12 ^b^	8.74 ± 0.09 ^c^
	PLA-nZH-Cu (3%)	-	3.83 ± 0.46 ^e^	8.98 ± 0.04 ^a^
Enterobacteriaceae	Control	1.74 ± 0.22	2.66 ± 0.37 ^b^	7.96 ± 0.19 ^a^
	PLA	-	2.64 ± 0.02 ^b^	7.78 ± 0.05 ^c^
	PLA-nZH (3%)	-	2.49 ± 0.33 ^c^	7.92 ± 0.04 ^a^
	PLA-nZH-Cu (1%)	-	2.85 ± 0.06 ^a^	7.87 ± 0.04 ^b^
	PLA-nZH-Cu (2%)	-	2.51 ± 0.18 ^c^	7.81 ± 0.34 ^c^
	PLA-nZH-Cu (3%)	-	2.14 ± 0.09 ^d^	7.52 ± 0.06 ^d^
*Salmonella* spp.	Control	Absence	Absence	Absence
	PLA	-	Absence	Absence
	PLA-nZH (3%)	-	Absence	Absence
	PLA-nZH-Cu (1%)	-	Absence	Absence
	PLA-nZH-Cu (2%)	-	Absence	Absence
	PLA-nZH-Cu (3%)	-	Absence	Absence

Values are presented as mean ± SD. *N* = 3. Values followed by different superscript letters in the same column indicate significant differences. - = Not analyzed on this day.

**Table 3 foods-11-00546-t003:** Instrumental color quality (chroma, hue, and E) of chicken breast meat in contact with PLA-nZH-Cu nanocomposite films during storage at 4 °C for 20 days.

Number of Days of Storage	Treatment	Instrumental Color		
		Chroma	Hue	E
0	Control	11.85 ± 2.87	1.39 ± 0.08	0.00 ± 0.00
10	Control	9.11 ± 1.44 ^a^	1.38 ± 0.01 ^a^	5.28 ± 3.52 ^b^
	PLA	10.29 ± 1.33 ^a^	1.28 ± 0.10 ^a^	8.59 ± 6.91 ^a^
	PLA-nZH (3%)	8.66 ± 2.08 ^a^	1.18 ± 0.14 ^a^	9.54 ± 4.96 ^a^
	PLA-nZH-Cu (1%)	7.99 ± 4.57 ^b^	1.20 ± 0.38 ^a^	6.24 ± 0.48 ^b^
	PLA-nZH-Cu (2%)	9.82 ± 1.13 ^a^	1.38 ± 0.18 ^a^	7.98 ± 3.89 ^b^
	PLA-nZH-Cu (3%)	7.94 ± 2.90 ^b^	0.62 ± 0.03 ^b^	9.75 ± 2.21 ^a^
20	Control	11.07 ± 2.77 ^a^	1.08 ± 0.05 ^a^	9.35 ± 3.96 ^b^
	PLA	12.76 ± 2.42 ^a^	1.23 ± 0.17 ^a^	8.62 ± 3.71 ^b^
	PLA-nZH (3%)	11.23 ± 0.40 ^a^	1.21 ± 0.23 ^a^	9.45 ± 3.66 ^b^
	PLA-nZH-Cu (1%)	9.15 ± 0.88 ^b^	1.13 ± 0.10 ^a^	10.03 ± 7.47 ^b^
	PLA-nZH-Cu (2%)	9.81 ± 0.28 ^b^	1.38 ± 0.14 ^a^	8.37 ± 4.35 ^b^
	PLA-nZH-Cu (3%)	11.70 ± 1.23 ^a^	1.09 ± 0.13 ^a^	12.15 ± 5.89 ^a^

Values are presented as mean ± SD. *N* = 3. Values followed by different superscript letters in the same column indicate significant differences.

**Table 4 foods-11-00546-t004:** Physicochemical quality (texture, weight, acidity, and pH) of chicken breast meat in contact with PLA-nZH-Cu nanocomposite films during storage at 4 °C for 20 days.

Number of Days of Storage	Treatment	Texture	Weight	Acidity and pH	
		Firmness (N)	Lost (%)	Lactic Acid (%)	pH
0	Control	8.53 ± 1.37	0.00 ± 0.00	0.13 ± 0.02	6.37 ± 0.32
10	Control	12.40 ± 3.04 ^b^	6.64 ± 1.96 ^b^	0.11 ± 0.02 ^a^	6.47 ± 0.33 ^a^
	PLA	21.74 ± 14.77 ^a^	12.32 ± 1.31 ^a^	0.09 ± 0.04 ^a^	6.54 ± 0.07 ^a^
	PLA-nZH (3%)	12.82 ± 5.58 ^b^	10.06 ± 1.30 ^a^	0.11 ± 0.01 ^a^	6.43 ± 0.04 ^a^
	PLA-nZH-Cu (1%)	20.18 ± 9.74 ^a^	10.45 ± 2.48 ^a^	0.13 ± 0.00 ^a^	6.15 ± 0.05 ^a^
	PLA-nZH-Cu (2%)	13.79 ± 2.67 ^b^	7.39 ± 0.69 ^b^	0.13 ± 0.00 ^a^	6.15 ± 0.06 ^a^
	PLA-nZH-Cu (3%)	13.09 ± 0.98 ^b^	9.10 ± 0.54 ^a^	0.11 ± 0.01 ^a^	6.41 ± 0.38 ^a^
20	Control	20.79 ± 1.55 ^b^	10.59 ± 3.89 ^b^	0.10 ± 0.02 ^a^	6.89 ± 0.08 ^a^
	PLA	19.98 ± 5.76 ^b^	18.64 ± 1.17 ^a^	0.07 ± 0.01 ^a^	7.60 ± 0.08 ^a^
	PLA-nZH (3%)	25.06 ± 15.13 ^a^	14.37 ± 1.72 ^b^	0.07 ± 0.01 ^a^	7.59 ± 0.12 ^a^
	PLA-nZH-Cu (1%)	21.06 ± 5.49 ^b^	12.10 ± 0.78 ^b^	0.07 ± 0.02 ^a^	7.57 ± 0.29 ^a^
	PLA-nZH-Cu (2%)	19.14 ± 3.93 ^b^	11.61 ± 1.42 ^b^	0.08 ± 0.01 ^a^	7.21± 0.18 ^a^
	PLA-nZH-Cu (3%)	17.21 ± 2.69 ^b^	10.96 ± 1.38 ^b^	0.08 ± 0.01 ^a^	7.56 ± 0.35 ^a^

Values are presented as mean ± SD. *N* = 3. Values followed by different superscript letters in the same column indicate significant differences.

## Data Availability

The data presented in this study are available from the corresponding author upon request.

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
