# Peer review of "Evaluation of the PLA-nZH-Cu Nanocomposite Film on the Micro-Biological, Organoleptic and Physicochemical Qualities of Packed Chicken Meat"

_foods, 2022, doi:10.3390/foods11040546_

Round 1

Reviewer 1 Report

The manuscript is straightforward. However, the following points should be considered for revision.

[1]. There is no in-depth discussion on how the nZH-Cu function as an antibacterial for the PLA film. What is the mechanism?

[2]. Lacking figures to demonstrate the physical appearance of the PLA film, the chicken meat packaging, etc.

[3]. The data presented only in table form could be easy to read, but it is less convincing if it is without figures as evidence.

[4]. Some of the data show that the performance of PLA is comparable to the PLA-nZH-Cu. Does it mean that unfilled PLA film also can be used for the packaging of chicken meat? An explanation should be given.

[5]. What is the significant difference between PLA-nZH and PLA-nZH-Cu? In some results, their properties are quite similar. What is the scientific reason?

Author Response

Response to Reviewer 1 Comments

Dear Reviewer,

The authors appreciate your valuable comments and suggestions, which help to improve our work. On the manuscript new version, modifications are yellow highlighted.

Point 1: There is no in-depth discussion on how the nZH-Cu function as an antibacterial for the PLA film. What is the mechanism?

Response 1: In the new version of the manuscript, in the Introduction section, there is an explanation of the mechanism of action of metallic copper nanoparticles and copper ions on bacteria (Page 3: line 92; page 4: lines 128-135). In the Results and Discussion section, there is also a discussion regarding the antibacterial action of nZH-Cu for PLA film (Page 10: lines 347-348; page 11: lines 349-355).

Point 2: Lacking figures to demonstrate the physical appearance of the PLA film, the chicken meat packaging, etc.

Response 2: The complete description of the development of nanocomposites belongs to the research of the current authors (Judith Vergara-Figueroa, Fabiola Cerda-Leal, Serguei Alejandro-Martín, and William Gacitúa). It is published in the article titled “Dual electrospinning of a nanocomposites biofilm: Potential use as an antimicrobial barrier” Materials Today Communications 25 (2020) 101671, https://doi.org/10.1016/j.mtcomm.2020.101671 [1]. This article is cited both in the original manuscript and in the new version of the manuscript (Page 4: lines 153-155).

In the current version of the manuscript, in the Materials and Methods section, we add a brief description and images of the manufacture of PLA films (Page 5: lines 164-170). In addition, we incorporate relevant images of the chicken meat pieces wrapped in PLA-nZH-Cu nanocomposite films, and the steps performed in the tests during the days of storage (Page 6: lines 196-201).

Point 3: The data presented only in table form could be easy to read, but it is less convincing if it is without figures as evidence.

Response 3: In the current version of the manuscript, we have added photographic records showing the materials used in each test (Page 6: lines 197-201, 218; page 7: lines 221-236; page 8: lines 239-242). This will help to better understand both the development of the research and the results presented in the tables.

Point 4: Some of the data show that the performance of PLA is comparable to the PLA-nZH-Cu. Does it mean that unfilled PLA film also can be used for the packaging of chicken meat? An explanation should be given.

Response 4: In the current version of the manuscript, a possible reason is explained why, in some results, pure PLA and PLA-nZH-Cu films present similar bacterial count values (Page 10: lines 303-330).

Point 5: What is the significant difference between PLA-nZH and PLA-nZH-Cu? In some results, their properties are quite similar. What is the scientific reason?

Response 5: PLA-nZH, corresponds to PLA films as a matrix, having nZH as disperse phase. The latter corresponds to zeolite nanoparticles with an increase in surface area by cleaning pores and channels. [1,2]. PLA-nZH-Cu corresponds to PLA films as a matrix, having zeolite nanoparticles with ion exchange with copper salt (Cu2+) as dispersed phase [1,2].

In the current version of the manuscript, Materials and Methods section, the difference between PLA-nZH and PLA-nZH-Cu is explained (Page 4: lines 152-155; page 5: lines 156-165).

In addition, in the Results and Discussion section, the reason why some bacteria count values present similar values is explained (Page 10: lines 303-330).

The description of the preparation of nZH and nZH-Cu is found in the Article “Obtaining Nanoparticles of Chilean Natural Zeolite and its Ion Exchange with Copper Salt (Cu2+) for Antibacterial Applications”, Materials 2019, 12, 2202; doi:10.3390/ma12132202 [2].

References

  1. Vergara-Figueroa, J.; Alejandro-Martin, S.; Cerda-Leal, F.; Gacitúa, W. Dual Electrospinning of a Nanocomposites Biofilm: Potential Use as an Antimicrobial Barrier. Mater. Today Commun. 2020, 25, doi:10.1016/j.mtcomm.2020.101671.
  2. Vergara-Figueroa, J.; Alejandro-Martín, S.; Pesenti, H.; Cerda, F.; Fernández-Pérez, A.; Gacitúa, W. Obtaining Nanoparticles of Chilean Natural Zeolite and Its Ion Exchange with Copper Salt (Cu 2+) for Antibacterial Applications. Materials (Basel). 2019, 12, 1–18, doi:10.3390/ma12132202.

Reviewer 2 Report

The work of Vergara-Figueroa et al. deals with the use of variable contents of nZH-Cu nanoparticles as an antimicrobial agent in order to improve the properties of the PLA matrix toward microbiological, organoleptic, and physicochemical acteristics of packed chicken breast meat. The study uses antimicrobial activities tests to assess the shelf life of food packaging.

Indeed, the paper is good written and the subject is interested for practical packaging applications. However, I propose the revision for this manuscript due to the following points:

  • The brittleness and heat distortion temperature are the major drawbacks of PLA and to overcome these drawbacks PLA polymer is blended with other biopolymers (i.e., PBAT, polycaprolactam, etc) to offer a convenient option for improving these properties. Thus, the authors could clarify how to fabricate PLA/nZn-Cu nanocomposite films to be wrapped film.
  • For packaging material, some tests and measurements shall be carried out, besides the antimicrobial activity, such as mechanical, WVTP and/or oxygen permeability, and thermal (DSC) properties. In this study, I recommend the authors to conduct the mechanical properties (i.e., tensile strength and elongation at break) for investigated films. Please visit this paper and cite it “Rational formulations of sustainable polyurethane/chitin/rosin composites reinforced with ZnO-doped-SiO2 nanoparticles for green packaging applications, Food Chemistry, 2022, 371, 131193”
  • The authors should make a comparison in table from literature survey between nZn-Cu nanoparticles and other synthetic or natural materials. For instance, “Development of dapsone-capped TiO2 hybrid nanocomposites and their effects on the UV radiation, mechanical, thermal properties and antibacterial activity of PVA bionanocomposites, Environmental Nanotechnology, Monitoring and Management, 2021, 16, 100482”, Antimicrobial low-density polyethylene/low-density polyethylene-grafted acrylic acid biocomposites based on rice bran with tea tree oil for food packaging applications, DOI:

10.1177/0892705720925140” and PLA/PBAT bionanocomposites with antimicrobial

natural rosin for green packaging, ACS Applied Materials and Interfaces, 2017, 9(23),

  1. 20132–20141
  • The authors must provide a figure/or photos to show the deterioration steps during 0, 10, and 20 days of storage.

Author Response

Response to Reviewer 2 Comments

Dear Reviewer,

The authors appreciate your valuable comments and suggestions, which help to improve our work. On the manuscript's latest version, modifications are yellow highlighted.

Point 1: The brittleness and heat distortion temperature are the major drawbacks of PLA and to overcome these drawbacks PLA polymer is blended with other biopolymers (i.e., PBAT, polycaprolactam, etc) to offer a convenient option for improving these properties. Thus, the authors could clarify how to fabricate PLA/nZH-Cu nanocomposite films to be wrapped film.

Response 1: Indeed, the PLA/nZH-Cu nanocomposite films were reinforced with acetylated cellulose nanofibers (acetylated CNF), to provide mechanical support. The complete description of the development of nanocomposites belongs to the research of the current authors (Judith Vergara-Figueroa, Fabiola Cerda-Leal, Serguei Alejandro-Martín, and William Gacitúa). It is published in the article entitled "Dual electrospinning of a nanocomposites biofilm: Potential use as an antimicrobial barrier" Materials Today Communications 25 (2020) 101671, https://doi.org/10.1016/j.mtcomm.2020.101671 [1]. This article is cited both in the original manuscript and in the latest version of the manuscript (Page 4: lines 152-155; page 5: lines 156-170).

For your information, I have written an excerpt from the description of the development of the manufacture of PLA/nZH-Cu nanocomposite films:

“Manufacture of biofilms from PLA: Two polymer blends were used to make the biofilms. One of the mixtures carried nZH-Cu as an antimicrobial agent, and the other mixture carried acetylated CNFs as reinforcement. Both mixtures were injected simultaneously using the electrospinning method in a dual configuration, and a rotary drum type collector was used to receive the sample [2–4]. The concentration of these elements was added according to the biopolymer amount (dry weight) in mixtures. For the elaboration of the biofilms, a General Factorial Design was used, with three replicas. Design-Expert 10.0.3 software was used [5]. For each polymer mixture, 10 ml of solution was prepared, 10 % of this solution corresponds to PLA and the remaining 90 % to solvent (chloroform: acetone 2:1 v/v) [6,7]. The best experimental conditions of electrospinning equipment reported by Gaitán and Gacitúa (2018), was conducted in this study due to the similarity of the equipment and type of PLA used. Distance from the tip of the injector to the collector of 20 cm, manufacturing voltage of 24 kV, the flow of the solution was estimated at 0.1 mL/hr [7]”.

Point 2: For packaging material, some tests and measurements shall be carried out, besides the antimicrobial activity, such as mechanical, WVTP and/or oxygen permeability, and thermal (DSC) properties. In this study, I recommend the authors conduct the mechanical properties (i.e., tensile strength and elongation at break) for investigated films. Please visit this paper and cite it Rational formulations of sustainable polyurethane/chitin/rosin composites reinforced with ZnO-doped-SiO2 nanoparticles for green packaging applications”, Food Chemistry, 2022, 371, 131193.

Response 2: Regarding the mechanical properties, I can mention that the PLA/nZH-Cu nanocomposite films were subjected to different tests, both mechanical and characterization. These results also belong to the current authors, Vergara-Figueroa et al., and are registered in https://doi.org/10.1016/j.mtcomm.2020.101671 [1]. The following conclusions are found in the aforementioned article: “Biofilms were developed based on the PLA polymer with the insertion of acetylated CNF and nZH-Cu, using the electrospinning technique. These nanoparticles were deposited on and between the fibers that make up the biofilms. SEM and morphological analyzes revealed a successful biofilm manufacturing process made up of uniform, continuous, and randomly arranged microfibers. Furthermore, a homogeneous distribution of copper in the biofilms was verified by SEM-EDX. The addition of acetylated CNF to the biofilm mixture causes an increase in its mechanical properties, determined by mechanical tests. The addition of nanoparticles was found to enhance the resistance to thermal degradation of biofilms, as revealed by the TGA degradation profile. The WVT assay demonstrated that the insertion of nanoparticles does not affect the permeability of biofilms based on PLA-nanoparticles. They show adequate characteristics to withstand handling in the hands of the consumer. Additionally, its biodegradability characteristics would help reduce contamination problems caused by the accumulation of plastic waste”.

Following his recommendation, in the current version of the manuscript, the article is cited Rational formulations of sustainable polyurethane/chitin/rosin composites reinforced with ZnO-doped-SiO2 nanoparticles for green packaging applications”, Food Chemistry, 2022, 371, 131193 (Page 3: line 107).  

Point 3: The authors should make a comparison in table from literature survey between nZH-Cu nanoparticles and other synthetic or natural materials. For instance,

“Development of dapsone-capped TiO2 hybrid nanocomposites and their effects on the UV radiation, mechanical, thermal properties and antibacterial activity of PVA bionanocomposites”, Environmental Nanotechnology, Monitoring and Management, 2021, 16, 100482,

“Antimicrobial low-density polyethylene/low-density polyethylene-grafted acrylic acid biocomposites based on rice bran with tea tree oil for food packaging applications”, DOI: 10.1177/0892705720925140, and

PLA/PBAT bionanocomposites with antimicrobial natural rosin for green packaging”, ACS Applied Materials and Interfaces, 2017, 9(23), 20132–20141.

Response 3: Following your recommendation in the current version of the manuscript, we have added the three citations in the results section. Indeed, the articles were of significant help in arguing our results (Page 10: line 203-330).

Point 4: The authors must provide a figure/or photos to show the deterioration steps during 0, 10, and 20 days of storage.

Response 4: Following your recommendations, we have added the images that show the steps of deterioration during the 0, 10, and 20 days of storage (Page 6: line 197-201; page 7: line 221-223; page 8: line 239-242).

References

  1. Vergara-Figueroa, J.; Alejandro-Martin, S.; Cerda-Leal, F.; Gacitúa, W. Dual Electrospinning of a Nanocomposites Biofilm: Potential Use as an Antimicrobial Barrier. Mater. Today Commun. 2020, 25, doi:10.1016/j.mtcomm.2020.101671.
  2. Kayaci, F.; Umu, O.C.O.; Tekinay, T.; Uyar, T. Antibacterial Electrospun Poly(Lactic Acid) (PLA) Nanofibrous Webs Incorporating Triclosan/Cyclodextrin Inclusion Complexes. J. Agric. Food Chem 2013, 61, 3901−3908, doi:10.1021/jf400440b.
  3. Toncheva, A.; Paneva, D.; Manolova, N.; Rashkov, I.; Mita, L.; Crispi, S.; Gustavo, D. Dual vs . Single Spinneret Electrospinning for the Preparation of Dual Drug Containing Non-Woven Fibrous Materials. Colloids Surfaces A Physicochem. Eng. Asp. 2013, 439, 176–183, doi:10.1016/j.colsurfa.2012.11.056.
  4. Ifkovits, J.; Sundararaghavan, H.; Burdick, J. Electrospinning Fibrous Polymer Scaffolds for Tissue Engineering and Cell Culture. J Vis Exp. NIH 2010, 27, 590–609, doi:10.1016/j.humov.2008.02.015.Changes.
  5. StatEase Design-Expert 2019.
  6. Buschle-Diller, G.; Cooper, J.; Xie, Z.; Wu, Y.; Waldrup, J.; Ren, X. Release of Antibiotics from Electrospun Bicomponent Fibers. Cellulose 2007, 14, 553–562, doi:10.1007/s10570-007-9183-3.
  7. Gaitán, A.; Gacitúa, W. Morphological and Mechanical Characterization of Electrospun Polylactic Acid and Microcrystalline Cellulose. BioResources 2018, 13, 3659–3673, doi:10.15376/biores.13.2.3659-3673.

Round 2

Reviewer 2 Report

The author's answer has resolved my doubts, and it is recommended to receive it.